# Qualitative Analysis of Multiple Phytochemical Compounds in Tojapride Based on UHPLC Q-Exactive Orbitrap Mass Spectrometry

**DOI:** 10.3390/molecules27196639

**Published:** 2022-10-06

**Authors:** Liying Zhang, Shihan Qin, Sunv Tang, Shuai E, Kailin Li, Jing Li, Wei Cai, Lei Sun, Hui Li

**Affiliations:** 1Institute of Digestive Diseases, Xiyuan Hospital of China Academy of Chinese Medical Sciences, Beijing 100000, China; 2School of Pharmaceutical Sciences, Hunan University of Medicine, Huaihua 418000, China; 3School of Pharmacy, Weifang Medical University, Weifang 261000, China; 4National Institutes for Food and Drug Control, Beijing 100050, China

**Keywords:** Tojapride, phytochemical compounds, UHPLC Q-Exactive Orbitrap mass spectrometry

## Abstract

Tojapride is composed of Caulis Perillae, Rhizoma Cyperi, Radix Glycyrrhizae, *Citrus aurantium* L., *Coptis chinensis* Franch, Pericarpium Citri Reticulatae, *Reynoutria japonica* Houtt, *Tetradium ruticarpum*, and *Cleistocactus sepium*. It has the effects of inhibiting gastric acid and relieving pain. It is clinically used for treating gastroesophageal reflux disease. To further study the pharmacodynamic properties of Tojapride, the systematic characterization of the chemical constituents in Tojapride was investigated using ultra-performance liquid chromatography with Q-Exactive Orbitrap mass spectrometry combined with parallel reaction monitoring for the first time. Eventually, a total of 222 compounds, including flavonoids, alkaloids, and glycyrrhizic acid derivatives, were identified based on the chromatographic retention times, MS/MS^2^ information, and bibliography data; a total of 218 of these were reported for the first time as being present in Tojapride. This newly developed approach provides a powerful tool for extending our understanding of chemical constituents of Tojapride, which can be further extended to other TCMP composition research.

## 1. Introduction

Tojapride is a Chinese medicine preparation composed of Caulis Perillae, Rhizoma Cyperi, Radix Glycyrrhizae, *Citrus aurantium* L., *Coptis chinensis* Franch, Pericarpium Citri Reticulatae, *Reynoutria japonica* Houtt, *Tetradium ruticarpum*, and *Cleistocactus sepium*. Tojapride is used for improving symptoms such as heartburn, chest pain, acid reflux, and nausea, and it is mainly used for treating gastroesophageal reflux disease (GERD), which is one of the most common chronic, progressive upper gastrointestinal tract disorders of the esophagus, characterized by heartburn and regurgitation symptoms [1,2]. It is reported that all of the single components in Tojapride have good pharmacological activities, including antibacterial, anti-inflammatory, and anti-tumor activities, and its use plays an important role in the treatment of GERD. However, according to the existing literature, there are few reports on the chemical composition of Tojapride. Thus, it is necessary to establish a valid method for systematically identifying the constituents of Tojapride, which will be beneficial to gain a deeper understanding of its pharmacodynamic properties of Tojapride.

Traditional Chinese medicine preparation (TCMP) has a long history in medical practice and health care, particularly in Asian and African countries. TCMP has attracted considerable attention in great many fields due to its effective therapeutic performance and low toxicity [3,4]. With the development of analytical techniques, the combination of ultra-high-performance liquid chromatography and mass spectrometry (UHPLC/MS) has had a significant impact on TCMP development in recent years [5,6]. They feature powerful analytical figures of merit (sensitivity, selectivity, speed of analysis) and have a wide scope of application in the qualitative and quantitative analysis of the TCMP [7].

In this research, an effective method using UHPLC with Q-Exactive Orbitrap mass spectrometry was established to characterize the chemical constituents of Tojapride. A total of 222 compounds were identified based on MS/MS^2^ data confirmation, retention time, and structural speculation; a total of 218 compounds were reported for the first time as being present in Tojapride. The results lay a foundation for the quality control of this medicine in the clinical use of Tojapride in the future. Additionally, LC-MS/MS has been proven to be an effective method for identifying TCMPs, and it also provides a new platform for qualitative analysis in many fields.

## 2. Results and Discussion

### 2.1. Analytical Strategy

The purpose of this study was to systematically identify the chemical constituents of Tojapride. Hence, an ideal strategy based on UHPLC-Q-Exactive Orbitrap MS combined with parallel reaction monitoring (PRM) was established. Firstly, the compound in Tojapride was extracted and enriched by ultrasonic extraction with 70% methanol. Secondly, the sample was injected into an UHPLC-Q-Exactive Orbitrap Mass Spectrometer to gain high-resolution MS data for Tojapride. Thirdly, the MS^2^ data for the trace ingredients in Tojapride were collected by UHPLC-Q-Exactive Orbitrap MS combined with PRM scanning. Lastly, the additional candidate chemicals were identified based on comparison with standards, summary DFIS, and neutral loss, as well as through comparison with the literature. 

### 2.2. Profiling of the Chemical Composition of Tojapride by LC-MS/MS

In total, 222 compounds were identified in Tojapride, including 133 flavonoids, and 39 glycyrrhizic acid derivatives, 22 alkaloids, and 28 others. Among them, 40 compounds were accurately identified by the reference standard. LC-MS/MS data regarding the chemical compound of Tojapride according to the LC-MS/MS analysis are presented in Appendix A. The high-resolution extracted ion flow diagram of Tojapride is shown in Figure 1.

#### 2.2.1. Characterization of the flavonoids in Tojapride

Compounds **28**, **36**, **42**, **53**, **55**, **60**, **63**, **64**, **79**, **81**, **101**, **111**, **112**, **122**, **123**, **127**, **131**, **140, 188**, **174**, **183**, **211**, and **212** were identified as orientin, liquiritin, vitexin, hyperoside, rutin, isoquercitrin, luteolin 7-O-β-D-glucoside, kaempferol 3-O-neohesperidoside, nicotiflorin, hesperidin, eriodictyol, baicalin, isoliquiritin, calycosin, quercetin, luteolin, naringenin, aloe emodin, nobiletin, wogonin, oroxylin A, proanthocyanidins, and emodin, respectively, by comparing the retention times and MS/MS data with the reference standards.

Compounds **51** and **80** possessed the same quasi-molecular ion and characteristic fragment ions as luteolin 7-O-β-D-glucoside. Thus, they were characterized as being luteolin 7-O-β-D-glucoside isomers. Likewise, compounds **32**, **49**, **56**, **71**, **84**, **86**, and **133** were deemed to be the isomers of orientin, hyperoside, vitexin, kaempferol 3-O-neohesperidoside, quercetin, nicotiflorin, and luteolin, respectively.

Compounds **17** and **27**, found at 8.34 and 10.60 min, respectively, possessed the same quasi-molecular ion [M−H]^−^ at *m*/*z* 771.1989, and their main fragment ions [M-H]^−^ were at *m*/*z* 609.1465, attributed to the neutral loss of the glucose reside (162 Da), and at *m*/*z* 463.0854, attributed to the neutral loss of a glucose + rhamnose moiety (308 Da). This further gave rise to product ions at *m*/*z* 301.0359 and 300.0273. Then, they were tentatively characterized as quercetin-O-glucoside-rutinoside. Similarly, compounds **94** and **99** were characterized as methyl-quercetin-O-rutinoside [5].

Compounds **20**, **24**, **33**, **41**, and **52** were eluted at 9.24, 10.48, 11.27, 11.93, and 12.80 min, respectively, and possessed the same quasi-molecular ion [M-H]^−^ at *m*/*z* 593.1512, and fragment ions at *m*/*z* 353.0676 [M-H-240]^−^, 383.0773 [M-H-210]^−^, and 473.1092 [M-H-120]. Therefore, they were identified as being vicenin 2 isomers [8].

Compounds **25**, **29**, and **43**, eluted at 10.48, 10.89, and 12.11 min, respectively, had the same quasi-molecular ion [M-H]^−^ at *m*/*z* 609.1461, and the fragmentation ion [M-H]^-^ at *m*/*z* 285.0406, corresponding to the loss of a lactose moiety (324 Da). Therefore, they were determined to be luteolin-O-lactoside. Similarly, compound 66 was confirmed as luteolin 7-O-glucuronide [9].

Compound **37** was eluted at 11.70 min, possessing the same quasi-molecular ions [M-H]^−^ at *m*/*z* 389.1242 and was deduced as being polydatin [10]. The main daughter ion at *m*/*z* 227.0711 was attributed to the loss of a glucose residue (162 Da). 

Compounds **46** and **120** possessed a molecular ion [M-H]^−^ at *m*/*z* 255.0663 (C_15_H_11_O_4_), and the main fragment ions [M-H]^−^ at *m*/*z* 153.0181, 135.0074, and 119.0488, consistent with the reference [11]. Thus, compounds **46** and **120** were identified as being liquiritigenin isomers. 

Compounds **35**, **44**, **110**, and **115** were eluted at 11.50, 12.16, 17.98, and 18.50 min, respectively, with the same quasi-molecular ion [M-H]^−^ at *m*/*z* 549.1614 (C_26_H_29_O_13_), and fragment ions [M-H]^−^ at *m*/*z* 255.0663, attributed to the neutral loss of glucose + arabinose (294 Da). In addition, the characteristic fragment ions at *m*/*z* 153.0181, 135.0074, and 119.0480 indicated that they possessed a liquiritigenin moiety. Thus, they were identified as being liquiritin apioside isomers [12]. Likewise, compounds **105** and **145** were deduced as being 6’-acetylliquiritin isomers [13].

Compounds **68** and **73**, detected at 13.76 and 14.23 min, respectively, possessed the same quasi-molecular ions [M-H]^−^ at *m*/*z* 433.1140, and the typical daughter ions at *m*/*z* 271.0616, obtained by the loss of a glucose residue (162 Da). However, the characteristic fragment ions at *m*/*z* 151.0028 and 119.0492 indicated that they possessed a naringenin group. Therefore, they were tentatively identified as being naringenin 7-O-glucoside isomers [5]. Likewise, compounds **22**, **26**, **40**, **47**, and **72** were tentatively proposed as being naringenin 4`-O-glucoside 7-O-rutinoside isomers, and compounds **65**, **76**, and **113** were tentatively identified as being narirutin isomers [14].

Compounds **87** and **137**, detected at 15.63 and 22.32 min, respectively, possessed the same quasi-molecular ions [M-H]^−^ at *m*/*z* 301.0718, and the main daughter ion at *m*/*z* 286.0484 and 151.0027. They were deduced as being hesperetin isomers compared with the previous literature [15]. Compounds **38**, **58**, **62**, **69**, **85**, and **93** with the same quasi-molecular ions [M-H]^−^ at *m*/*z* 771.2353 were eluted at 11.76, 13.18, 13.42, 13.88, 15.52, and 16.20 min, respectively. The main MS/MS fragment ion at *m*/*z* 301.0721, formed by the neutral loss of glucose + rutinose moiety (470 Da), further produced the characteristic fragment ions at *m*/*z* 286.0482 and 151.0027, indicating the presence of hesperetin. Therefore, they have been deduced as being hesperidin 7-O-glucoside isomers [15]. Likewise, compound **88** was found to be hesperetin 7-O-β-D-glucuronide [15].

Compounds **89**, **107**, **134**, and **146** were found at 15.63, 17.46, 21.11, and 24.04 min, respectively, which showed a common precursor ion [M-H]^−^ at *m*/*z* 431.0984. The major fragment ion at *m*/*z* 269.0459, due to the loss of a glucose reside (162 Da), further gave rise to the product ions at *m*/*z* 241.0458 and 225.0555. Therefore, they were tentatively characterized as being aloe emodin 8-glucoside isomers [16].

Compounds **92**, **95**, **143**, **148**, and **156** appeared at retention times of 16.20, 16.57, 23.69, 24.27, and 25.64 min, respectively, possessing the same quasi-molecular ions [M-H]^−^ at *m*/*z* 445.1140. The main fragment ions at *m*/*z* 283.0613 were generated based on the neutral loss of a glucose reside (162 Da), and further produced the ion at *m*/*z* 268.0383, and was identified as oroxylin A-O-glucoside. Likewise, compound **14****4** was found to be an oroxylin A derivative. 

Compounds **119** and **175** were eluted at 19.81 and 27.71 min, respectively, and yielded a deprotonated ion [M-H]^−^ at *m*/*z* 285.0768 and fragment ions at *m*/*z* 164.0106 and 151.0028. Based on these MS data, they were suggested as being sakuranetin isomers [17]. Likewise, compounds **125** and **130** were characterized as being poncirin isomers [18]. Furthermore, compounds **185**, **193**, **204**, and **217** were tentatively identified as the isomers of 6,7,8,3’,4’-Pentamethoxyflavanone, 3,5,6,7,8,3’,4’-Heptamethoxyflavone, licoricone, and licoisoflavone B, respectively [11,12,19]. 

Compounds **150**, **161**, **164**, **182**, **208**, and **213** were observed at 24.46, 25.85, 26.17, 28.39, 31.61, and 32.25 min, respectively, possessing the same quasi-molecular ions [M+H]^+^ at *m*/*z* 389.1231; the fragment ion at *m*/*z* 359.0756 generated by the loss of 2CH_3_ group (30 Da) further gave rise to the product ions at *m*/*z* 169.0130 and 197.0080. Thus, they were found to be 3-Hydroxy-5,6,7,8,4’-Pentamethoxyflavanone isomers [19]. 

Compounds **155**, **163**, and **218** with the same quasi-molecular ions [M+H]^+^ at *m*/*z* 359.1125 were eluted at 25.48, 26.06, and 33.15 min and identified as being 5-Hydroxy-3,6,7,8-Tetramethoxyflavone isomers [19]. The main MS^2^ fragment ion at *m*/*z* 329.0653 was due to the loss of the 2CH_3_ residue (30 Da).

Compounds **169**, **177**, **184**, **190**, 201, and **203** were detected at 27.17, 27.90, 28.74, 29.50, 30.31, and 30.86 min, respectively, with the same empirical molecular formula C_20_H_20_O_7_, matched to that of the isosinensetin isomers [19]. The fragment ions at *m*/*z* 358.1043 and 343.0810, generated by the sequential loss of a CH_3_ group (15 Da) in MS^2^, further gave rise to the product ions at *m*/*z* 315.0862 [M-2CH_3_-CO]^+^. 

Compounds **141**, **153**, and **167** were found at 23.36, 25.10, and 26.64 min, respectively, possessing the same quasi-molecular ions [M+H]^+^ at *m*/*z* 419.1337, and were tentatively identified as being 3-Hydroxy-5,6,7,8,3’,4’-Hexamethoxyflavone isomers [19]. The main product ions at *m*/*z* 404.1089 and 389.0863 were attributed to the successive loss of a CH_3_ radical (15 Da), and the obtained product ions at *m*/*z* 371.0762 were attributed to the loss of H_2_O from the product ions at *m*/*z* 389.0863.

Compounds **189** and **194** were eluted at 29.15 and 29.73 min, respectively, possessing the same quasi-molecular ions [M+H]^+^ at *m*/*z* 343.1176. The main daughter ion at *m*/*z* 313.0701 was attributed to the loss of a 2CH_3_ radical (30 Da). The fragment ions were obtained at *m*/*z* 285.0754 due to the loss of a CO radical (28 Da), and they were deduced as being 5,7,3’,4’-Tetramethoxyflavone isomers [19].

#### 2.2.2. Characterization of the Alkaloids in Tojapride

Compound **31** was detected at 11.08 min, possessing the quasi-molecular ions [M+H]^+^ at *m*/*z* 505.2306 and MS/MS fragment ion at *m*/*z* 342.1698, due to the neutral loss of glucose reside (162 Da), and this indicated the presence of magnoflorine. Thus, it was identified as magnoflorine-O-glucoside [5]. 

Compound **21** was eluted at 9.54 min, possessing the quasi-molecular ions [M+H]^+^ at *m*/*z* 342.1705, and was identified as magnoflorine by comparing the retention times and MS and MS^2^ information with those of the standards. Compounds **38**, **49**, and **57**, which possessed the same quasi-molecular ions and characteristic fragment ions as magnoflorine, were characterized as being magnoflorine isomers.

Compounds **74**, **82**, and **90**, detected at 14.30, 15.27, and 15.85 min, respectively, possessed the same quasi-molecular ions [M+H]^+^ at *m*/*z* 324.1236 and the main fragment ion at *m*/*z* 309.0993 and 294.0759, generated by the sequential loss of the CH_3_ group (15 Da); they were identified as demethyleneberberine [20].

Compounds **77**, **98**, and **116** had the quasi-molecular ions [M+H]^+^ at *m*/*z* 322.1079, and yielded the fragment ions at *m*/*z* 307.0837 due to the neutral loss of a CH_3_ (15 Da). The product ions at *m*/*z* 279.0887 were obtained by the loss of CO (28 Da) moieties from the main ion at *m*/*z* 307.0837. Hence, they were deduced as being groenlandicine [20].

Compounds **83**, **102**, **106**, and **114** were eluted at 15.27, 16.93, 17.22, and 18.40 min, respectively, possessing the same quasi-molecular ions [M+H]^+^ at *m*/*z* 338.1392, and were deduced as being columbamine [20]. The main daughter ion at *m*/*z* 322.1072 and 308.0915 was due to loss of a OH radical (16 Da) and 2CH_3_ radical (30 Da) in MS^2^, respectively. Similarly, compounds **117** and **126**, with the same precursor ions at *m*/*z* 352.1550, were characterized as palmatine [20] and had an MS/MS fragment ion at *m*/*z* 336.1226 and 308.1277, attributing to the loss of an OH radical (16 Da) and 2CH_3_ radical (30 Da), respectively.

Compounds **100** and **108** were eluted at 16.90 and 17.53 min, respectively, and they yielded the parent ion [M+H]^+^ at *m*/*z* 320.0923; they were deduced as being coptisine, according to the MS and MS/MS spectra [20]. 

Compounds **104**, **121**, and **129** were eluted at 17.16, 19.47, and 20.37 min, respectively, having the same precursor ion [M+H]^+^ at *m*/*z* 336.1240, and the fragment ion at *m*/*z* 321.0994 and 320.0914, generated by the loss of the CH_3_ residue (15 Da) and CH_3_-H residue (16 Da), respectively. This further gave rise to the product ions at *m*/*z* 292.0964 [M-CH_3_-H-CO]^+^, these compounds were tentatively proposed as being berberine isomers [20].

#### 2.2.3. Characterization of the Glycyrrhizic Acid Derivative in Tojapride

Compounds **34** and **158**, which appeared at a retention time of 11.40 and 25.77 min, respectively, possessed the same quasi-molecular ion [M-H]^−^ at *m*/*z* 471.2013; they were identified as being a limonin isomer. 

Compound **186** was eluted at 28.96 min, with the deprotonation ion [M-H]^-^ at *m*/*z* 821.3965 and was unambiguously identified as glycyrrhizic acid by comparing the reference standards; the parent ion yielded the main fragment ions *m*/*z* 351.0562 (C_19_H_11_O_7_), 193.0348 (C_6_H_9_O_7_), and 113.0232 (C_5_H_5_O_3_) by the loss of C_23_H_51_O_9,_ C_36_H_53_O_9,_ C_37_H_57_O_13,_ respectively. These fragment ions can be used as DFIS to identify glycyrrhizic acid and its derivatives.

Compounds **157**, **192**, **198**, **202**, and **210**, found at 25.69, 29.57, 29.88, 30.81, and 31.83 min, respectively, possessed the same quasi-molecular ions and characteristic fragment ions as glycyrrhizic acid, and were characterized as being glycyrrhizic acid isomers [12].

Compounds **135**, **138**, **139**, **142**, **149**, and **154** were detected at 21.83, 22.49, 22.96, 21.83, 22.49, and 22.96 min, respectively, and they possessed the same quasi-molecular ion [M-H]^-^ at *m*/*z* 853.3863, and the main characteristic fragment ions [M-H]^−^ at *m*/*z* 351.0562 (C_19_H_11_O_7_), 193.0348 (C_6_H_9_O_7_), and 113.0232 (C_5_H_5_O_3_). Hence, they were identified as being 22-hydroxyl-licorice saponin G2 isomers [21]. 

Compounds **147**, **152**, **162**, 166, **171**, **176**, **180**, and **187**, observed at 24.16, 24.91, 26.04, 26.54, 27.36, 27.77, 28.29, and 29.08 min, respectively, possessed the quasi-molecular ion [M-H]^-^ at *m*/*z* 837.3914, and generated the diagnostic fragment ions at (*m*/*z* 351.0562, 193.0348, and 113.0232). Thus, they were identified as licorice-saponin G2 isomers [11]. Likewise, compounds **160** and **168** were tentatively proposed as being licorice-saponin A3, and licorice saponin E2 isomers, respectively [12].

Compounds **151**, **199**, **205**, **209**, and **214** at *m*/*z* 823.4122, had the same molecular formula C_42_H_64_O_16_ and appeared at a retention time (tR) of 24.65, 30.20, 31.16, 31.75, and 32.56 min, respectively. They were suggested as being licorice saponin J2 isomers [22], 

Compounds **178**, **191**, **196**, and **200** were eluted at 28.00, 29.51, 29.83, and 30.23 min, respectively, possessing the same quasi-molecular ions [M-H]^−^ at *m*/*z* 863.4071 and deduced as being 22 β-acetoxylglycyrrhaldehyde isomers by comparison with the existing literature. Similarly, compounds **165** and **173** were tentatively characterized as being a 22-acetoxyglycyrrhizin isomer [22]; compounds **206**, **216**, and **219** were deduced as being a licorice saponin C2 isomer [22].

#### 2.2.4. Other Chemical Constituents in Tojapride

Compounds **1**, **2**, **4**, **5**, **8**, **9**, **11**, **14**, **15**, **18**, **23**, **54**, **59**, **96**, and **222** were found at 0.88, 0.94, 1.46, 1.46, 2.42, 3.03, 4.24, 6.29, 6.37, 8.80, 10.37, 12.95, 13.18, 16.60, and 40.04 min, respectively, corresponding to quinic acid, malic acid, citraconic acid, gallic acid, protocatechuic acid, chlorogenic acid, methyl gallate, cryptochlorogenic acid, caffeic acid, octanedioic acid, salicylic acid, isoferulic acid, azelaic acid, abscisic acid, and oleanic acid, respectively, by comparing the retention time and MS^2^ data with reference standards.

Compounds **3** and **7** were eluted at 1.29 and 2.05 min, respectively, with the quasi-molecular ion [M-H]^−^ at *m*/*z* 331.0671 and the fragmentation ion at *m*/*z* 169.0134, corresponding to the loss of one glucose moiety (162 Da). The product ions at *m*/*z* 169.0133 and 125.0232 indicated the presence of gallic acid. Therefore, compounds **3** and **7** were determined to be a 1-galloyl-β-glucose isomer [4]. 

Compound **6** possessed a molecular ion [M-H]^−^ at *m*/*z* 197.0455 and fragment ions [M-H]^−^ at *m*/*z* 179.0343 and 152.8941, and it was preliminarily identified as being alpha-(3,4-dihydroxyphenyl) lactic acid by comparison with MS^2^ in the literature [23]. Likewise, compounds **170** and **197** were tentatively identified as being glycycoumarin isomers, and compound **215** was a glycyrol isomer [11]. 

Compounds **10** and **12** were eluted at 4.03 and 5.57 min, and they possessed a similar molecular ion [M-H]^−^ at *m*/*z* 153.0193 and fragment ion at *m*/*z* 109.0283 to protocatechuic acid. Accordingly, compound **135** was identified as a protocatechuic acid isomer. Likewise, compounds **13**, **16**, and **19** were identified as being an isoferulic acid isomers; compounds **45**, **70**, and **136** were characterized as being abscisic acid isomers. 

### 2.3. Pharmacological Activity of Chemical Ingredients in the Tojapride

The compounds identified in this study included flavonoids, alkaloids, glycyrrhizic acid derivative, and other compounds. Flavonoids are widely distributed in various medicinal plants and are the effective components of many traditional Chinese medicines. According to reports, flavonoids have been repeatedly studied because of their favorable pharmacological activity. For example, as emodin, naringin, and neohesperidin can be used in the treatment of GI motility disorders, they may play an important role in the pharmacodynamic properties of Tojapride, which itself plays a role in pain relief, reduction in acid regurgitation, and the promotion of GI motility [24,25]. The anti-inflammatory activity of Tojapride was confirmed many years ago, and it has been widely used in patients with non-erosive reflux disease. Most of the compounds identified in this study, such as quercetin, magnoflorine, and glycyrrhizic acid, have anti-inflammatory activity, which may be the material basis for the treatment of GERD.

## 3. Experimental

### 3.1. Materials and Chemicals

The chemical reagents (HPLC-grade), including acetonitrile and methanol, were purchased from Fisher Scientific (New Jersey, USA). The LC–MS-grade formic acid was acquired from Thermo Fisher Scientific Co., Ltd. (Carlsbad, CA, USA). The deionized water was prepared using a Milli-Q system (Bedford, MA, USA). Other solvents were of an analytical grade. Detailed information regarding the 40 standards is listed in Appendix A. Tojapride was provided by the Xiyuan Hospital, China Academy of Chinese Medical Sciences (Beijing, China). 

### 3.2. Preparation of Standard and Sample Solutions 

A Tojapride sample (1 mg) was dissolved in 70% methanol/water (10 mL) in a 15 mL tube and vortexed (5 min). Subsequently, the sample was transferred into the ultrasonic machine for 0.5 h. Then, the solution was centrifuged at 15,000 RCF for 15 min and the supernatant was injected into an HPLC vial. The sample solution (3 μL) was then injected into the LC-MS system for further analysis.

The 40 chemical standards (1 mg) were accurately weighed and dissolved in methanol to obtain the stock standard solutions. The mixed working solution was prepared by adding 10 µL of the stock standard solution into a 1 mL volumetric flask.

### 3.3. Instruments and LC–MS/MS Conditions

Qualitative analyses were performed using a UHPLC-Q-Orbitrap Mass Spectrometer (Thermo Fisher Scientific, Carlsbad, CA, USA). The UHPLC separation was carried out using a Thermo Scientific Hypersil GOLD™ aQ (100 × 2.1 mm, 1.9 μm). The mobile phase consisted of 0.1% formic acid, water (A), and acetonitrile (B), with a flow rate of 0.3 mL/min under the gradient program of 95 to 5% (A) for an initial 2 min, 95–5% (A) from 2 to 5 min, 90–10% (A) from 5 to 20 min, 75–25% (A) from 20 to 30 min, 55–45% (A) from 30 to 40 min, 20–80% (A) from 40 to 45 min, followed by 5% (A) from 45 to 50 min.

Mass spectrometric analysis was operated in negative ion and positive ion modes using an ESI, and the full scan mass spectral data were collected over a range from *m*/*z* 100 to 1400 at a resolution of 35,000 and targeted MS^2^ at a resolution of 17,500, triggered by parallel reaction monitoring (PRM). The optimum source parameters were as follows: spray voltage (−3.2 kV); spray voltage (+3.5 kV), the sheath gas flow rate (35 arb); aux gas flow rate (10 arb); capillary temperature (320 °C); heater temperature (350 °C); S-lens RF level (60); and the stepped, normalized collision energies (20%, 40%, and 60%). 

### 3.4. Data Analysis

The UHPLC-MS data in Tojapride was processed using Xcalibur software version 4.2; the minimum peak intensity was set at 10,000. The chemical formulae for all parent and fragment ions of the selected peaks were calculated from the accurate mass using a formula predictor by setting the parameters as follows: C [0–50], H [0–70], O [0–30], and N [0–5]; the mass tolerance of MS and MS2 was within 5 ppm, respectively

### 3.5. Establishment of Diagnostic Fragment Ion (**DFIs**)

The compounds in the same category possess identical carbon skeletons that will generate similar characteristic fragment ions. In this study, the fragment ion patterns of flavonols, flavanones, and glycyrrhizic acid were investigated using LC-MS/MS. The fragmentation pathway of flavonols and flavanones is shown in Figure 2A,B. The same fragmentation ions were identified as 151.002 (C_7_H_3_O_4_), 107.012 (C_6_H_3_O_2_), and 119.050 (C_8_H_7_O), which could be considered the DFIS of flavonols and flavanones. Likewise, the DFIS values (351.056, C_19_H_11_O_7_; 193.035, C_6_H_9_O_7_; and 113.023, C_5_H_5_O_3_) of glycyrrhizic acid derivatives were displayed in Figure 2C.

## 4. Conclusion

A useful approach using UHPLC-Q-Exactive Orbitrap MS combined with the PRM technique was established in the present study as an effective tool for the assessment of the chemical composition of Tojapride. As a result, a total of 222 compounds were identified and 218 were isolated for the first time from Tojapride. The methods are simple, rapid, and sensitive, and provide useful MS/MS^2^ data. The research results not only extend our understanding of the chemical constituents of Tojapride in the existing study, but also exhibit a wide application for the characterization and profiling of compounds in different samples.

## Figures and Tables

**Figure 1 molecules-27-06639-f001:**
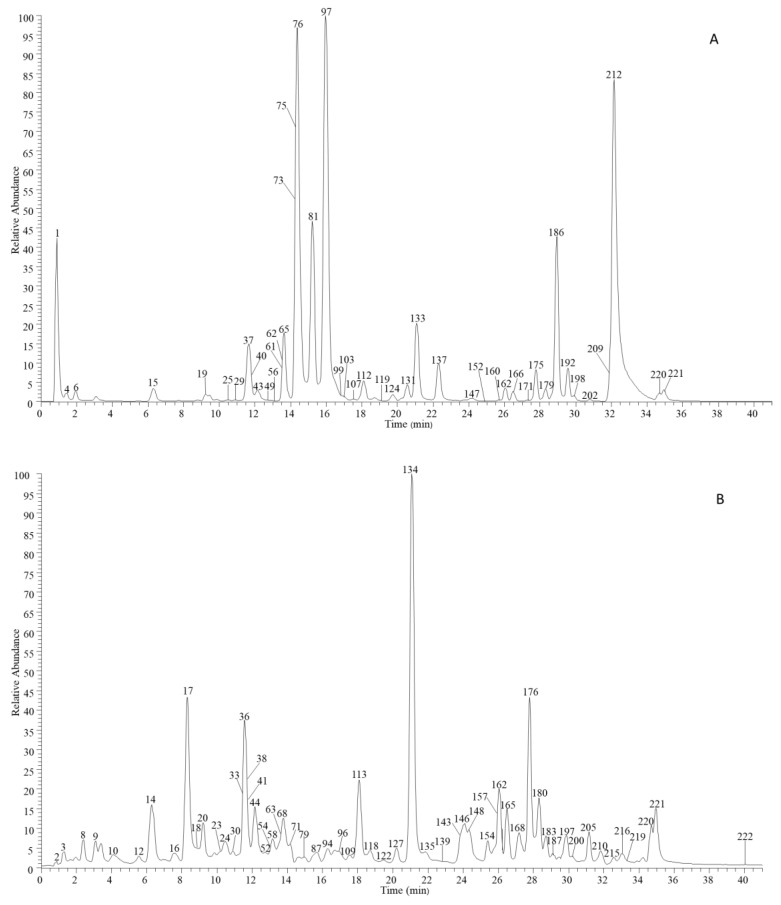
The high-resolution extracted ion flow diagram of Tojapride (**A**): 133.0142, 169.0142, 191.0561, 197.0455, 269.0455, 271.0612, 283.0612, 301.0718, 353.0878, 389.1242, 417.1191, 431.0984, 433.114, 463.1246, 549.1614, 579.1719, 593.1512, 593.1876, 609.1825, 621.1825, 741.2248, 821.3965, 837.3914, 879.4020; (**B**): 137.0244, 153.0193, 173.0819, 179.0350, 183.0300, 187.0976, 193.0506, 255.0663, 263.1288, 283.0612, 285.0405, 287.0561, 301.0354, 331.0671, 351.0874, 353.0878, 365.1031, 367.1187, 381.1344, 417.1191, 431.0984, 433.114, 445.0776, 445.114, 447.0933, 455.3531, 459.1297, 463.0882, 549.1614, 593.1301, 593.1512, 609.1461, 621.1825, 623.1618, 741.2248, 771.1989, 771.2353, 805.4016, 819.3809, 823.4122, 837.3914, 853.3863, 863.4071, 879.402, 983.4493; (**C**): 137.0244, 169.0142, 187.0976, 193.0506, 255.0663, 263.1288, 269.0455, 271.0612, 283.0612, 285.0768, 287.0561, 301.0354, 331.0671, 367.1187, 381.1344, 431.0984, 433.114, 445.114, 447.0933, 459.1297, 463.0882, 463.1246, 549.1614, 593.1301, 593.1512, 593.1876, 609.1461, 609.1825, 621.1825, 623.1618, 741.2248, 771.1989, 771.2353, 805.4016, 821.3965, 823.4122, 837.3914, 853.3863, 863.4071, 879.4020; (**D**): 153.0193, 173.0819, 179.0350, 183.0300, 187.0976, 255.0663, 263.1288, 287.0561, 301.0354, 331.0671, 351.0874, 353.0878, 365.1031, 367.1187, 445.1140, 447.0933, 455.3531, 459.1297, 463.0882, 593.1301, 593.1512, 609.1461, 621.1825, 623.1618, 741.2248, 771.1989, 771.2353, 805.4016, 819.3809, 823.4122, 853.3863, 863.4071, 879.4020, 983.4493; (**E**): 343.1176, 359.1125, 375.1074, 389.1231, 419.1337, 463.0871, 465.1391, 471.2013, 505.2306; (**F**): 320.0923, 322.1079, 324.1236, 336.1240, 338.1392, 342.1705, 352.1550, 373.1282, 403.1387, 433.1493. (**A**–**D**) EIC in negative mode; (**E**,**F**) EIC in positive mode.

**Figure 2 molecules-27-06639-f002:**
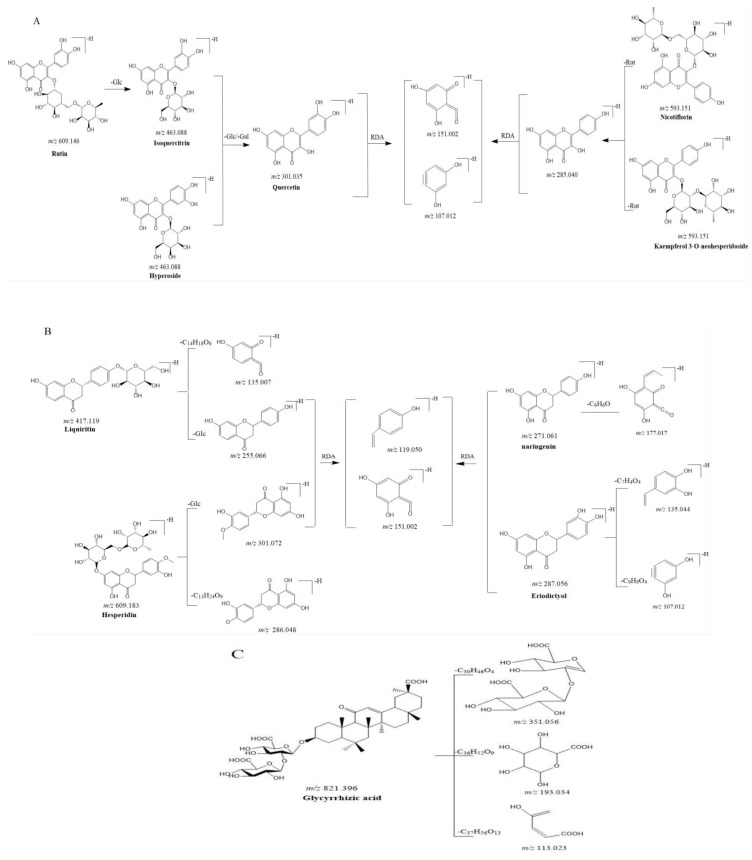
Proposed selected fragmentation pattern of components identified in Tojapride: Flavonols (**A**), flavanones (**B**), glycyrrhizic acid (**C**).

## Data Availability

Not applicable.

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
