# Peer review of "Qualitative Analysis of Multiple Phytochemical Compounds in Tojapride Based on UHPLC Q-Exactive Orbitrap Mass Spectrometry"

_molecules, 2022, doi:10.3390/molecules27196639_

Round 1

Reviewer 1 Report

I greatly appreciate the author’s initiative to study the Tojapride herb due to its significant role in the medicinal world. Mass spectrometry is a great tool to understand the chemical nature and structure of complicated naturally existing or synthetic chemical compounds.

The authors can greatly improve this article by reviewing the article for typing and grammatical errors.

Line 57. 70% of what?

Section: Characterization of compounds

Please review the following suggestions:

I understand that the authors performed a very detailed analysis of plant extract and successfully curated useful information from it using mass spectrometry. However, I would like to make a comment about completing the story by looping all the compounds back to their medicinal significance. There is very little information about this in the article, and it can significantly improve the article if incorporated precisely.

Please consider reviewing the article for grammatical and typing errors.

This article is of great importance as it provides scientific evidence of the phytochemical properties of Tojapride plants in clinical science. I encourage you to make suggested changes to it and resubmit it.

Author Response

Reviewer 1:

I greatly appreciate the author’s initiative to study the Tojapride herb due to its significant role in the medicinal world. Mass spectrometry is a great tool to understand the chemical nature and structure of complicated naturally existing or synthetic chemical compounds.

  1. The authors can greatly improve this article by reviewing the article for typing and grammatical errors.

Answer:Thank you for your valuable and thoughtful comments. It has been revised according to the comments of reviewers.

  1. Line 57. 70% of what?

Answer:Thank you for your valuable and thoughtful comments. It is 70% methanol/water, which was revised in line 66 according the reviewer’s suggestion.

  1. I understand that the authors performed a very detailed analysis of plant extract and successfully curated useful information from it using mass spectrometry. However, I would like to make a comment about completing the story by looping all the compounds back to their medicinal significance. There is very little information about this in the article, and it can significantly improve the article if incorporated precisely.

Answer:Thank you for your valuable and thoughtful comments. The medicinal significance was revised in the section 2.4 Pharmacological activity of chemical ingredients in the Tojapride

Reviewer 2 Report

Title: Qualitative analysis of multiple phytochemical compounds in Tojapride based on UHPLC Q-Exactive Orbitrap Mass Spectrometry.

The authors have selected an interesting topic. Tojapride is currently used as a standard, commercial, patented medicine at the China Academy of Chinese Medical Sciences Xiyuan Hospital. Tojapride significantly reduces acid reflux and heartburn symptoms in individuals with GERD. Tojapride contains several different anti-inflammatory Chinese herbs.

The design and presentation of the study findings is acceptable.

However, there are some suggestions to be followed:

1. The abstract must be elaborated to highlight the novel findings.

2. Line 26-27: Include the complete composition of Tojapride (avoid using "etc")

3. The introduction is very short. The authors must shed some light on the biological activities reported on Tojapride.

4. Table 1. Retention times and mass spectral data of Tojapride. This table can be presented as a supplementary file, as this disturbs the flow of the manuscript.

5. Revise the English language. 

Author Response

Reviewer 2:

The authors have selected an interesting topic. Tojapride is currently used as a standard, commercial, patented medicine at the China Academy of Chinese Medical Sciences Xiyuan Hospital. Tojapride significantly reduces acid reflux and heartburn symptoms in individuals with GERD. Tojapride contains several different anti-inflammatory Chinese herbs.The design and presentation of the study findings is acceptable. However, there are some suggestions to be followed:

  1. The abstract must be elaborated to highlight the novel findings.

Answer:Thank you for your valuable and thoughtful comments.  It has been revised according the reviewer’s suggestion.

  1. Line 26-27: Include the complete composition of Tojapride (avoid using "etc")

Answer:Thank you for your valuable and thoughtful comments. It has been revised according to the comments of reviewers.

  1. The introduction is very short. The authors must shed some light on the biological activities reported on Tojapride.

Answer:Thank you for your valuable and thoughtful comments.

the biological activities reported of Tojapride has been added in line 39-41 according the reviewer’s suggestion.

  1. Table 1. Retention times and mass spectral data of Tojapride. This table can be presented as a supplementary file, as this disturbs the flow of the manuscript.

Answer:Thank you for your valuable and thoughtful comments. the Table 1 of the article was revised according the reviewer’s suggestion.

5.Revise the English language.

Answer:Thank you for your valuable and thoughtful comments. It has been revised according to the comments of reviewers.

Figure 1. the certification of language editing
